# Pharmacogenomics and the Management of Mood Disorders—A Review

**DOI:** 10.3390/jpm13071183

**Published:** 2023-07-24

**Authors:** Kristian Kleine Schaars, Roos van Westrhenen

**Affiliations:** 1Outpatient Clinic Pharmacogenetics, Parnassia Psychiatric Institute/PsyQ, Overschiestraat 57, 1062 HN Amsterdam, The Netherlands; 2Institute of Psychiatry, Psychology & Neuroscience (IoPPN), King’s College London, De Crespigny Park, Denmark Hill, London SES 8AF, UK; 3Department of Psychiatry & Neuropsychology, Faculty of Health, Medicine and Life Sciences, Maastricht University, P.O. Box 6161, 6229 ER Maastricht, The Netherlands

**Keywords:** pharmacogenomics, pharmacogenetics, metabolizer status, phenotype, dosing guidelines, mood disorders

## Abstract

Due to the chronic relapsing nature of mental disorders and increased life expectancy, the societal burden of these non-communicable diseases will increase even further. Treatments for mental disorders, such as depression, are available, but their effect is limited due to patients’ (genetic) heterogeneity, low treatment compliance and frequent side effects. In general, only one-third of the patients respond to treatment. Today, medication selection in psychiatry relies on a trial-and-error approach based mainly on physicians’ experience. Pharmacogenetic (PGx) testing can help in this process by determining the person-specific genetic factors that may predict clinical response and side effects associated with genetic variants that impact drug-metabolizing enzymes, drug transporters or drug targets. PGxis a discipline that investigates genetic factors that affect the absorption, metabolism, and transport of drugs, thereby affecting therapy outcome. These genetic factors can, among other things, lead to differences in the activity of enzymes that metabolize drugs. Studies in depressed patients show that genotyping of drug-metabolizing enzymes can increase the effectiveness of treatment, which could benefit millions of patients worldwide. This review highlights these studies, gives recommendations and provides future perspectives on how to proceed with PGx testing. Finally, it is recommended to consider genotyping for *CYP2D6* and *CYP2C19*, when there is an indication (side effects or inefficacy).

## 1. Introduction

Treatment resistant depression (TRD) is typically described as the occurrence of an inadequate response after an adequate treatment with antidepressant agents (in terms of dose, duration, and compliance), among patients suffering from a depressive disorder [1]. The European Union’s Committee for Human Proprietary Medicinal Products (CHMP) defines TRD as follows: ‘A patient is considered therapy-resistant when consecutive treatments with two products of different classes, used for a sufficient length of time at an adequate dose, fail to induce an acceptable effect’ [1].

Effective treatments for mental disorders are available, but their effect is limited due to patients’ (genetic) heterogeneity and poor treatment compliance due to frequent adverse events. Only one-third of the patients respond to treatment and experience remission, and lots of patients experience side effects when using antidepressant medications [2,3]. Studies have shown that psychiatry relies on a trial-and-error approach that combines physicians’ experience with clinical indicators [4,5]. Pharmacogenetic testing can help in this process by determining the person-specific genetic factors that may predict clinical response and side effects associated with genetic variants that impact drug-metabolizing enzymes, drug transporters or drug targets [4,5,6].

Pharmacogenetics is a discipline that investigates genetic factors that affect the absorption, metabolism, and transport of drugs, thereby affecting therapy outcome. These genetic factors can, among other things, lead to differences in the activity of enzymes that metabolize drugs. 

Genetic variants which are less than 1% present in the population are referred to as mutations and more than 1% as genetic polymorphisms [6]. For several polymorphisms in genes encoding for metabolizing enzymes, the effect on the activity of the enzyme has been established. Besides genetic polymorphisms, also other (non-genetic) factors can influence enzyme activity, such as co-medication, smoking, diet and diseases [5]. The genetic composition of the genes, referred to as the ‘genotype’, is translated into a predicted ‘phenotype’ [6] The only pharmacogenes that have been shown to have a clear relationship with clinical outcome so far have been the cytochrome P450 genes and in particular *CYP2C19* and *CYP2D6* [7]. Several research groups have published articles on the genotype of metabolizing enzymes and their predicted phenotype [8,9]. 

Traditional classification usually divides patients into four groups: (1) Poor metabolizers are homozygous carriers of two loss-of-function (*CYP2C19*Null/Null or *CYP2D6*Null/Null, also sometimes referred to as *CYP2C19*PM/PM or *CYP2D6*PM/PM) alleles, they do not possess active enzymes, and therefore, their specific enzymatic capacity is completely abolished; (2) Intermediate metabolizers carry genotypes that are causing substantially reduced, but not absent, enzyme capacity; (3) Normal metabolizers homozygously carry Wt alleles and are associated with reference or expected enzymatic capacity. Normal metabolizers may also carry other genotypes as long as the enzymatic capacity is not significantly different compared with Wt/Wt carriers; (4) Ultrarapid metabolizers carry genotypes connected with significantly increased enzymatic capacity compared with Wt/Wt carriers [6].

In general, patients with little enzyme activity experience more side effects of medication caused by an accumulation of the medicine in the body, while patients with a higher enzyme activity experience less effectiveness of medication. There are some exceptions to this rule. An example of this is the drug venlafaxine of which the metabolite is just as effective as the parent compound. However, it is indicated that side-effects are still greater in PM [10]. Another example is that of medicines taken as a pro-drug such as fluoxetine which are metabolized into an active compound. The effects of PGx testing on pro-drugs is not yet well understood [11]. Variants in cytochrome enzyme *CYP2D6* are known to have the poor metabolizer (PM) phenotype in 5 to 10% of the population with a Northwestern European background [12]. These individuals have little or no *CYP2D6* enzyme activity, dependent on their allele type. They have an increased risk of side effects of medication that is metabolized by this particular enzyme, due to a slower degradation of certain drugs (for example with nortriptyline), and therefore higher blood drug concentrations. There also may be undertreatment if the drug has to be converted by *CYP2D6* to the active metabolite, as is the case with some opioids (e.g., tramadol). Genetic polymorphisms influencing drug metabolism are common. Besides the poor metabolizer (PM) status, intermediate metabolizer status (IM: decreased enzyme activity) and ultrarapid metabolizer status (UM: increased enzyme activity) may predispose patients for an increased risk on both side effects and therapeutic ineffectiveness. For example, in the (non-Finnish) European population, 3% is *CYP2C19* PM and7% is *CYP2D6* PM. In addition, in this population 27% is *CYP2C19* IM and 40% *CYP2D6* IM [6,12]. In other populations the proportion of non-NM can be even higher. In, for example, the East-Asian population, 15% is *CYP2C19* PM and 47% is *CYP2C19* IM. So far, there is a lack of large randomized clinical trials in non-Caucasian populations.

### 1.1. Potential Benefit of Pharmacogenetics in Treatment Resistant Depression

Antidepressants in general aim to increase monoaminergic neurotransmission by blocking monoamine reuptake. However, these effects are neither necessary nor sufficient for treatment response and the effectiveness of therapy is therefore less than optimal [6]. Despite intensive effort in neuroscience research, very few new psychopharmacological agents have entered the market in recent decades. Therefore, since molecular targets for psychiatric drugs are not yet fully elucidated, and many of the currently available drugs will likely remain the cornerstone of pharmacotherapy in psychiatry for the foreseeable future, it is of paramount value to maximize their effectiveness [6]. The current selection of an appropriate antidepressant drug to a great extent still relies on psychiatrists’ clinical experience as well as on a potentially long trial and error approach, with potential serious adverse consequences that may include, for example, untreated suicidal ideation and behaviors [6,12,13,14]. 

### 1.2. Dosing Guidelines

Since the share of drug metabolism catalyzed by *CYP2C19*, *CYP2D6*, and other enzymes is not uniform across psychiatric drugs, when it comes to meaningful guidelines, each drug needs to be considered separately. In 2005, the Dutch Pharmacogenetic Working Group (DPWG) formulated dosing advice based on pharmacogenetic metabolizer status (www.Kennisbank.knmp.nl, accessed on 1 June 2023); updated guidelines from DPWG and/or the Clinical Pharmacogenetics Implementation Consortium (CPIC) are provided (see www.Pharmgkb.org, accessed on 1 June 2023). Although such advice has been accepted worldwide with subtle differences, it is nonetheless still rarely used in clinical settings [6].

Recently a Dutch guideline “Pharmacogenetics in daily Psychiatric Clinical Practice" was drafted as assigned by the Dutch Psychiatric Association (NVvP) with funding from the Dutch Royal Medical Association (KNMG) [7]. The main goal of this guideline was to provide guidance on how genotyping could be used to contribute to the best clinical psychiatric practice. The purpose was to address basic questions such as when to genotype, how to request genotyping, which genes to investigate and how to interpret the genotype results. Patients can also order genotyping via commercial companies, but this usually results in the outcome of more genes, for which not always sufficient evidence is present to be used in clinical practice. This guideline provides a description of dosing advice to be used in clinical psychiatric practice, based on current scientific evidence available. Because currently, insufficient data are available on the effects of genes involved in the pharmacodynamics of psychotropic drugs, these were excluded but will be included in the updates to this guideline when more knowledge about the clinical applicability becomes available [15]. In the Netherlands, there is quite a unique situation because of the fact that 16 hospital pharmacies offer non-commercial pharmacogenetic testing. These tests are reimbursed by Dutch health insurance companies. The DPWG and the infrastructure in this small country make for good working alliances and cooperation between professionals from different centers in different areas and with worldwide collaborations. With this, to our knowledge, first clinical psychiatric guideline, drawn up by clinicians working together with other professionals, a start can be made by implementing pharmacogenetics in clinical psychiatry at a larger scale. Through alignment with other initiatives as e.g., the CPIC and the recently started pan-European PSY-PGx research consortium that was set-up by van Westrhenen, an international effort could be undertaken to establish a worldwide clinical guideline for implementation of pharmacogenetics in psychiatry, which go beyond the already available more theoretical guidelines. In several countries across the world there are pharmacogenetics outpatient clinics, for example in the US and Canada and the Netherlands. To our knowledge the only non-commercial clinic, which delivers reimbursed care is the one in the Netherlands. Currently, around the world, more non-commercial outpatient clinics are being setup.

## 2. Clinical Pharmacogenomic Studies in Psychiatry

Earlier, a systematic review was conducted and a clinical guideline on implementation was published [7]. In short, all prospective clinical studies in psychiatric patients investigating cytochrome P450 enzymes and reporting clinical outcome were assessed with validated questionnaires and they were compared with standard care in patients using antidepressants using the GRADE method (earlier described in van Westrhenen et al. Frontiers 2021 [7]). The literature search was updated in May 2023 with the same search terms. To date fourteen prospective clinical studies have been published on the effect of genotyping compared to standard care (not further specified) in mood disorder patients [16,17,18,19,20,21,22] (Table 1). A total of eleven studies describe the results of randomized studies [18,19,20,21,22,23,24,25,26,27,28]. The studies by Hall-Flavin [16,17] describe the results of prospective observational cohort studies and the study by Oslin and colleagues describes the result of an open-label controlled trial [29]. All fourteen studies were targeted at depressed patients, only the study by Bradly et al. also included patients with anxiety disorder [21]. One study was found investigating the use of genotyping in psychotic patients in comparison with standard care. 

Six studies used the Genesight test [16,17,18,22,24,29], two studies used NeuroPharmagen [20,23] another two studies used TaqMan probe–PCR and mass array [28,29], one study used the CNSDose [19], another used Genecept [28], one used the NeuroIDgenetix test [21] and one study did not report on the used test [25].

The Genesight test used was a combination test for allele variations in five genes: *CYP1A2*, *CYP2D6*, *CYP2C19*, the serotonin transporter gene (*SLC6A4*), and serotonin receptor 2A (*HTR2A/5-HTR2A*).

Hall-Flavin was the first to examine whether the use of pharmacogenetic information (Genesight test) yielded gains in the clinical outcome of depressed patients (MDD; Major depressive disorder) in a prospective pilot study (Genesight-I study) [16]. Adults aged 25–75 years with psychiatrist-diagnosed MDD and a minimum score of 14 on the Hamilton Depression Scale (HAMD-17) were included. The study was sponsored by the industry (AssureRx Inc., Mason, OH, USA) and two authors were still employed by the Mayo Clinic at the time of design of the study but joined AssureRx during the course of the study. After 8 weeks, the decrease in severity of depressive disorder was greater in the group of patients treated based on pharmacogenetic advice (30.8% decrease in HAMD-17 score) compared to the group of patients receiving standard care (18.2% decrease in HAMD-17 score, *p* = 0.04) [16].

**Table 1 jpm-13-01183-t001:** Prospective RCT’s comparing PGx guided vs. non-guided pharmacotherapy (TAU).

Study Design	Genotyping Method	Number of Patients	Outcome
Open label, prospective cohort [16]	*CYP1A2*, *CYP2D6*, *CYP2C19* (Genesight)	n = 25 Genotyping guided vs. n = 26 TAU	Genotyping leads to more reduction in depression scores
Open label, prospective cohort [17]	*CYP1A2*, *CYP2D6* *CYP2C19* (Genesight)	n = 114 Genotyping guided vs. n = 113 TAU	Genotyping leads to more reduction in depression symptoms
RCT, double-blind [18]	*CYP1A2*, *2C9*, *2C19*, *2D6*, *SCL6A4*, *HTR2A* (Genesight)	n = 26 guided vs. n = 25 TAU	Genotyping results in higher response and remission rates
RCT, double-blind [19]	*CYP2D6*, *CYP2C19*, *ABCB1*, not otherwise specified	n = 74 Genotyping guided vs. n = 74 TAU	Genotyping 2.52 times more likely to remit
RCT, double-blind [20]	*CYP2D6* and others, not specified (NeuroPharmagen)	n = 155 guided vs. n = 161 TAU	Genotyping results in higher response rate and better tolerability
RCT, double blind, both depressed and anxiety patients [21]	*CYP1A2*, *2C9*, *2C19*, *2D6*, *3A4*, *3A5*, *SCL6A4*, *COMT*, *HTR2A*, *MHFR* (NeurolDgenetix)	n = 352 Genotyping guided vs. n = 333 TAU	Genotyping leads to higher response rates and remission rates in patients with depression or anxiety
RCT, double blind [22]	*CYP1A2*, *2C9*, *2C19*, *2B6*, *2D6*, *HTR2A*, *SCL6A4*	n= 1167 in total	Genotyping leads to higher response and remission rates in depressed patients
RCT, double blind [23]	*CYP2D6* and others, not specified (NeuroPharmagen)	n = 52 genotyping guided vs. n = 48 TAU	Genotyping leads to more reduction in depression scores and higher response rates
RCT, double blind [24]	*CYP1A2*, *CYP2B6*, *CYP2C9*, *CYP2C19*, *CYP2D6*, *CYP3A4*, *HTR2A*, and *SLC6A4* (Genesight) and *MC4R*, *CNR1*, *NPY*, *GCG*, *HCRTR2*, *NDUFS1* (in enhanced Genesight)	n = 90 genotyping guided, n = 93 enhanced genotyping guided vs. n = 93 TAU	Genotyping did not lead to a higher decrease in depressive symptoms or higher response or remission
RCT, double blind [25]	*CYP2D6* and *CYP2C19* (not otherwise specified)	n = 56 Genotyping guided vs. n = 55 TAU	Pharmacogenomic testing was associated with a faster achievement of therapeutic drug concentrations and a decrease in adverse reactions, but not with a decrease in depressive symptoms
RCT, double blind [26]	*CYP2C19*, *CYP2D6*, *CYP1A2*, *SLC6A4*, and *HTR2A* (TaqMan probe–PCR and mass array)	n = 31 Genotyping guided vs. n = 40 TAU	Genotyping might not considerably improve the clinical efficiency and safety
RCT, single blind [27]	*CYP1A2*, *CYP2B6*, *CYP2C9*, *CYP2C19*, *CYP2D6*, *CYP3A4*, *HLA*, *5HTTLPR*, *HTR2A*, *HTR2C*, *POLG*, *SLC6A4* and *UGT1A4* (TaqMan probe–PCR and mass array)	n = 55 Genotyping guided vs. n = 47 TAU	Genotyping did not lead to significant group differences although a trend was shown
RCT, double blind [28]	Genecept (7 CYP enzymes genes with 45 variants, 11 pharmacodynamic or other genes)	n = 151 Genotyping guided vs. n = 153 TAU	Pharmacogenomic testing was not associated with an improvement of symptoms
Open label, prospective cohort [29]	*CYP1A2*, *CYP2B6*, *CYP2C9*, *CYP2C19*, *CYP2D6*, *CYP3A4*, *UGT1A4*, *UGT2B15*, *SLC6A4*, *HTR2A*, *HLA-B*1502* and *HLA-B*1502* (Genesight)	n = 966 Genotyping guided vs. n = 978 TAU	Genotyping based medication advice reduced the number of prescribed drugs with a drug-gene interaction compared to usual care

Hall-Flavin conducted a second prospective cohort study with 227 depressed patients (Genesight-II study [17]. As the previous study by this group [10], the diagnosis was supported by a HAMD-17 score of at least 14. This time, the study involved patients between the ages of 18 and 72 with a depressive disorder. It was reported that after eight weeks there was a greater reduction in depressive symptoms when pharmacotherapy was prescribed using genotyping (intervention group: 46.9% decrease in score on the HAMD-17 versus 29.9% decrease in score on this questionnaire in the standard care group, *p* < 0.01). The response rate, defined as >50% decrease in score, was also higher in the intervention group than in the standard care group, 44.4% versus 23.7%, respectively, (odds ratio (OR) = 2.58 95% confidence interval (BI) 1.33 to 5.03). Similar results were found for the percentage of patients with remission (defined as a score of less than or equal to 7 on the HAMD-17), intervention group: 26.4%, standard care group: 12.9%; OR = 2.42, 95%BI 1.09 to 5.39, *p* =0.03 [17].

The third study using the Genesight panel in depressed patients was a double-blind randomized trial [18]. The clinical effect was measured with the HAMD-17 and the Quick Inventory of Depressive symptoms (QIDS-SR and QIDS-CR) for 10 weeks. All authors were employed by industry (AssureRx Health, Inc., Mason, OH, USA). The mean improvement in HAMD-17 scores at week 10 was higher in the group with a dosing recommendation based on genotyping (30.8% vs. 20.7%; *p* = 0.28). Response, which was defined as a 50% reduction in HAMD-17 scores, was higher in the Genesight arm compared to the usual treatment arm (36,0% and 20.8% respectively, OR = 2.14; 95% CI: 0.59–7.69). On top of this, remission at ten weeks, a HAMD-17 score < 7, was 20.0% in the Genesight arm compared to 8.3% in the TAU arm (OR = 2.14; 95% CI: 0.59–7.69). Together this means that both remission and response improved greatly in the genotyped group. Next to this, reduction in depressive symptoms was greater in the Genesight group compared to TAU (30.8% vs. 20.7%; *p* = 0.28). Lastly, TAU patients who started a contraindicated medication based on their genotype had very little improvement (only 0.8%) in depressive symptoms. This was a great contrast compared to the Genesight group who started on a contraindicated medication but had a 33.1% improvement (*p* = 0.06).

The fourth study, using the Genesight panel, included n = 1167 outpatients with depressive disorder with a score > 14 on the HAMD-17 [22] and with a patient- or clinician-reported inadequate response to at least one antidepressant. Clinical effect was measured at 0 (baseline), 4, 8, 12, and 24 weeks by a blinded central rater with HAMD-17 (primary), as well as with QIDS and Patient Health Questionnaire (PHQ). In this study there was no significant difference in symptom improvement at 8 weeks between the two treatment groups (27.2% guided versus 24.4% standard care, *p* = 0.107), but this was the case for response (≥50% decrease in HAMD-17) at week 8 (26.0% versus 19.9%, *p* = 0.013) and remission defined as HAMD-17 < 7 (15.3% versus 10.1%, *p* = 0.007) [16].

In another study, a test for polymorphisms of *CYP2D6*, *CYP2C19*, and ABCB1 (CNSDose) was used [19]. This 12-week, double-blind randomized study was industry-sponsored (n = 152). In half of the patients, the prescriber received dosage recommendations (intervention group). The HAMD-17 was measured after 12 weeks. There was a greater improvement in depressive symptoms after ten weeks: in the intervention group there was 30.8% improvement in HAMD-17 compared to 20.7% improvement in the standard care group, but this difference was not statistically significant (*p* = 0.28). No statistically significant differences in response rate and percentage of patients with remission were found either: Response rate OR = 2.14; 95%BI = 0.59 to 7.69. With the reported data (9 of 25 cases in the intervention group and 5 of 24 cases in the control group, one dropout in both groups) we arrive at a relative risk of 1.73 (95%BI = 0.68 to 4.42). For the percentage of patients with remission, an OR of 2.75 (95%BI = 0.48 to 15.80) was reported. With the reported data (5 of 25 cases in the intervention group and 2 of 24 cases in the control group), the relative risk can again be calculated: 2.4 (95%BI = 0.51 to 11.21). Remission is defined here as a final score on HAMD-17 of less than or equal to 7. The authors further reported that the intervention group had progressed on average 27.6% on the QIDSC-16 and the standard care group 22.1%, this difference was not statistically significant [19].

The next study used the Neuropharmagen panel which contained *CYP2D6* and others (not clearly specified). N = 155 patients were treated with genotype guided medication compared to n = 161 patients in the standard care group in a double-blind set-up [14]. The PGx-guided treatment group had a higher responder rate compared to standard care at 12 weeks (47.8% vs. 36.1%, *p* = 0.0476; OR = 1.62 [95%CI 1.00–2.61]), and this difference increased after removing subjects in the PGx-guided group when clinicians explicitly reported not to follow the test recommendations (51.3% vs. 36.1%, *p* = 0.0135; OR = 1.86 [95%CI 1.13–3.05]). Effects were more consistent in patients with one–three failed drug trials [20].

Bradley et al. investigated a total of n = 685 patients with depression and/or anxiety, of which n = 246 were diagnosed with depression, and n = 204 with both depression and anxiety (n = 450 with depression) in a prospective, randomized, subject and rater-blinded approach [21]. They assessed HAM-D17 and HAM-A scores at 0,4,8 and 12 weeks using the NeuroID genetix panel, investigating 10 genes including *CYP1A2*, *2C9*, *2C19*, *2D6*, *3A4*, *3A5*, *SCL6A4*, *COMT*, *HTR2A*, and *MHFR* [21]. It was described that in patients diagnosed with severe depression, response rates at 8 weeks were 55% in the experimental group versus 28% in the control group, and response rates (*p* = 0.001; OR: 4.72 [1.93–11.52]) and remission rates (*p* = 0.02; OR: 3.54 [1.27–9.88]) were significantly higher in the pharmacogenetics-guided group as compared to the control group at 12 weeks. In addition, when both moderate and severe patients were included in the analysis at 8 weeks comparing the experimental and control group the response rates were 49% and 41% respectively. At 12 weeks the response rates were significantly higher in the experimental group (*p* = 0.01, OR:2.03, 1.23–3.33) However, for mild depression patients, no significant effect of pharmacogenomic guided treatment was found. In addition, patients in the experimental group diagnosed with anxiety showed a meaningful improvement in HAM-A scores at both 8 and 12 weeks (*p* = 0.02 and 0.02, respectively), along with higher response rates (*p* = 0.04; OR: 1.76 [1.03–2.99]). From these results, they concluded that pharmacogenetic-guided medication selection significantly improves outcomes of patients diagnosed with severe depression or anxiety, in a variety of healthcare settings [21].

Han et al. conducted an 8-week randomized, rater and participant blinded, clinical trial [23]. N = 100 participants with MDD were included and divided into either the PGx or a TAU treatment arm. Treatment was based on either pharmacogenetic analysis using the Neuropharmagen testing panel or treatment left to the assessment of the own healthcare provider. Treatments were compared at 8 weeks using the HAM-D17 questionnaire. There was a significant mean change in HAMD score in favor of the PGx group with a –4.1-point difference at the end of the eighth week (−16.1 ± 6.8 −12.1 ± 8.2, *p* = 0.010). Similarly, response rates based on HAMD score were also significantly higher in the PGx group with 28.1% difference (*p* = 0.014). Remission rates were higher in the PGx group, although not significantly (*p* = 0.071). Additionally, adverse reactions as reason for drop-out was higher in the TAU group compared to the PGx group (n = nine in TAU compared to n = four in PGx). With these results they concluded that PGx may be a better option for treatment of MDD with respect to tolerability and effectiveness. 

A 52-week study was launched by Tiwari et al. In this participant and rater blinded three-arm RCT [24], patients were randomized into one of three groups. The first group consisted of patients where the medication prescribers received a standard pharmacogenomic test report (GEN) and could provide medication advice based on these results. In the second group medication prescribers received an enhanced pharmacogenomic test report (EGEN) or received advice to give the usual treatment (TAU). The pharmacogenomic test reports were based on the Genesight panel. The enhanced report consisted of additional genes which could give insight into anti-psychotic induced weight gain. The primary endpoint was symptom recovery (change in HAM-D17 score) at week 8 and secondary endpoints were remission and response rates at week 8. A total of n = 570 patients were planned for enrollment of which 276 were finally included in the per protocol analysis. There was no significant difference in the primary endpoints between the GEN and TAU arms (GEN 24.4 vs. TAU 22.6, *p* = 0.738) nor was there a significant difference in EGEN and GEN arms (*p* = 0.244). The guided treatment also did not prove significantly more effective in response (30.3% versus 22.7%), and remission rates (15.7% versus 8.3%) compared to TAU. However, the trial was underpowered to detect statistically significant differences.

One study looked at the effect of pharmacogenetics informed treatment on the use of tricyclic antidepressants (TCA) [25]. Vos and colleagues investigated and analyzed n = 111 patients from whom *CYP2D6* and *CYP2C19* genotypes were assessed. The primary outcome was days until attainment of a therapeutic TCA plasma concentration. They found that treatment based on pharmacogenetics led to a faster achievement of therapeutic concentrations as assessed by therapeutic drug monitoring (mean [SD], 17.3 [11.2] vs. 22.0 [10.2] days; *p* = 0.04). Next to this, (*p* = 0.001), severity (*p* = 0.008), and burden (*p* = 0.02) also decreased.

Another non-commercially funded study was performed by Shan and colleagues [26]. They investigated n = 71 participants between the ages of 18 and 51 with MDD as described by DSM-V with a HAM-D17 score of ≥17. Pharmacogenomic testing was performed by a non-commercial laboratory. Genomic DNA was isolated and analyzed using TaqMan probe–PCR and mass array which detects genetic polymorphisms including *CYP2C19*, *CYP2D6*, *CYP1A2*, *SLC6A4*, and *HTR2A*. Participants received either PGx based treatment or TAU. Analysis revealed no statistically significant difference (*p* > 0.05) in HAMD-17 reduction rates (60.68% guided versus 52.38% in unguided) as well as response rates, and remission rates between the PGx guided and unguided groups. There were also no significant differences in adverse effects incidence (55.56% guided and 57.89% unguided). Based on these findings, Shan et al. suggested that pharmacogenomic testing might not considerably improve the clinical efficiency and safety for the PGx guided group. 

Another study utilizing the TaqMan probe–PCR and mass array was undertaken by McCarthy et al. n = 102 patients with MDD were subjected to PGx guided treatment or TAU for 8 weeks [27]. Endpoints were measured using the Clinical Global Impressions Scale (CGI). Remission was also measured using CGI and was defined as any patient ending the trial with a CGI score of either 1 or 2. Subjects significantly improved over the course of the trial (*p* = 0.01), however between the treatment arms there was no significant effect, although it favored PGx guided treatment (*p* = 0.08). Correction for diagnosis revealed that patients with MDD (n = 49), recovered similarly overtime independent of treatment arm (*p* = 0.55). The authors concluded that there were no significant group differences although a trend was shown.

An 8-week multicenter, participant and rater-blinded randomized controlled trial, conducted by Perlis and colleagues, examined PGx guided treatment versus TAU using a Genecept assay kit [28]. The Genecept assay includes seven CYP enzymes genes with a total of 45 variants and 11 pharmacodynamic or other genes (no further specifications). Included participants had to be diagnosed with MDD with a minimum SIGH-D score of 18 and had a failure of treatment with at least one anti-depressant. A total of n = 296 participants yielded valuable outcomes. The primary outcome of interest was change in SIGH-D score. No significant difference was observed between the PGx and TAU groups at week 8 (*p* = 0.53). Likewise, no significant effect of guided treatment was detected on response rates (39.7% PGx vs. 48.0% TAU; −7.8% difference, *p* = 0.17) and remission rates (24.0% PGx vs. 30.7% TAU; −6.1% difference, *p* = 0.23). Based on these results, it was concluded that pharmacogenomic testing was not associated with an improvement of symptoms. 

Lastly, Oslin and colleagues also utilized the Genesight panel [29]. They initiated a 24-week non-commercially sponsored open label trial. In this study they investigated remission rates and the risks for receiving an antidepressant with drug-gene interactions as measured by the Patient Health Questionnaire-9 (PHQ-9), comparing a PGx guided group versus a TAU group. A total of n = 1944 patients between 18 and 80 years with an MDD diagnosis were included in the final analysis. The results indicated that the risk of a predicted drug-gene interaction was lower in the PGx guided group compared to the TAU group (*p* < 0.001, no gene interaction vs. moderate/substantial interaction). At 24 weeks there was no significant effect of treatment on remission observed in the PGx versus TAU groups (*p* = 0.45). Oslin et al. concluded that providing PGx test results reduced the number of prescribed drugs with a drug-gene interaction compared to usual care.

### 2.1. Side Effects

One study examined the effectiveness of genotyping versus standard care with respect to adverse events measured by the prespecified relevant outcome measures [23]. Adverse reactions as reason for drop-out was higher in the TAU group compared to the PGx group (n = nine in TAU compared to n = four in PGx) in the trial conducted by Han and colleagues. 

### 2.2. Therapeutic Drug Monitoring (TDM)

Only one study was found comparing the effect of treatment based on pharmacogenetics with standard care in psychiatric patients taking antidepressants while analyzing therapeutic plasma concentrations [29]. With the use of TDM, Vos and colleagues found out that genotyping led to a faster achievement of therapeutic TCA plasma concentrations compared to TAU. 

For certain drugs, such as TCAs, lithium, and clozapine, as well as haloperidol, the German Laborgemeinschaft für Neuropsychopharmakologie und Pharmacopsychiatrie (AGNP) recommends TDM in their guidelines. However, TDM is not required by default for all psychopharmaceuticals. According to the latest update of the German guideline on TDM, TDM is not necessary for all SSRIs and SNRIs since large interindividual differences in drug concentrations in the blood result in the absence of a reliable therapeutic window [15]. Monographs are also available at www.tdm-monografie.org, accessed on 1 June 2023. However, it should be realized that the recommendations regarding TDM for psychopharmaceuticals are mostly based on pharmacokinetic (PK) data. RCTs have rarely been conducted in which the added value of TDM in clinical practice has been demonstrated in terms of clinical outcomes measured with correct measurement scales. In patient care, it is more common to make recommendations based on available theoretical knowledge without information from good RCTs and thus clinically relevant scientific evidence. 

After starting treatment with TCAs, lithium, and clozapine, TDM can be used to further adjust the dosage [15]. TDM is of added value when using these drugs, because here the risk of relapse is higher at low exposures to the drug, or to prevent toxicity at higher levels (especially for TCAs). For lithium, clozapine and TCAs such as nortriptyline, imipramine and amitriptyline, target concentrations are well defined. If TCAs are prescribed for an indication other than depressive or anxiety disorder, such as for pain in general practice, this is at a lower dose and TDM makes little sense [7].

It therefore seems useful to consider genotyping if there are adverse events or abnormal kinetics during treatment with a TCA based on TDM (for example, strikingly high levels at low doses or low levels at high doses). Genotyping is a way to explain these discrepancies with regard to the levels found, so that this can be taken into account in the future when adjusting dosages or starting other drugs [7]. Lithium is excreted by the kidneys and therefore no CYP enzymes are involved [7].

Although the AGNP Consensus Guideline also recommends TDM for haloperidol, it is currently not the case that TDM for haloperidol is widely used and usually it is only determined on indication. For antipsychotics, TDM is not considered necessary by default, except for clozapine for which a reliable therapeutic window exists. TDM is not considered necessary by default for most SSRIs and SNRIs, due to large interindividual differences in drug concentrations in the blood and the consequent lack of a reliable therapeutic window [7]. Exceptions are cases where adherence to therapy is suspected [7].

## 3. Evidence from the Literature: Summary

It can be concluded that: 

1. There is some evidence from fourteen prospective studies that pharmacogenetics-guided treatment with antidepressants leads to a greater likelihood of remission of depressive symptoms and better treatment response than standard care.

2. One study was found comparing the effect of pharmacogenetics-based treatment on the occurrence of adverse events with standard care in psychiatric patients taking or about to take antidepressants. The risk for adverse events in standard care seems to be higher, however, the evidence is scarce and non-significant.

3. One study was found comparing the effect of treatment based on pharmacogenetics with standard care in psychiatric patients who are taking antipsychotics. This was a study conducted by Jürgens and colleagues investigating the effects of genotyping on antipsychotic drug persistence, i.e., how long a patient would stay on a described drug [30]. They investigated n = 528 patients which were allocated into one of three treatment arms. In the end, the results did not support the use of genotyping in the treatment of psychotic patients. However, no clinical scale (i.e., the PANSS) was used as a study outcome and there is a lack of clarity in which genotyping kit and drugs were used. Since not all antipsychotics benefit from genotyping, indicating which antipsychotics are used is crucial.

### 3.1. Pharmacogenetic Recommendations for Psychiatric Practice

Below the main recommendations for clinical use for pharmacogenetics in psychiatric practice from the Dutch guideline are summarized:

### 3.2. General Recommendations for All Psychopharmaca

When considering genotyping, inform the patient and involve the patient in order to achieve shared decision making with regard to genotype.When genotype information is already available at the time of the prescription, use this information to select the right drug and the right dose for the right patient.Consider genotyping, when there is an indication (side effects or inefficacy of *CYP2D6* and *CYP2C19*. Preemptive genotyping is therefore not recommended yet for psychotropic drugs.For some psychotropic drugs determination of *CYP1A2*, *CYP2C9* and/or *CYP3A4* can be of added value, but only determine those after consultation of a clinical pharmacologist or PharmD with pharmacogenetic expertise.Ensure that available genotyping results are recorded in the (electronic) patient file and that this information is shared with medication prescribers (such as the GP) as well as the pharmacy.Consult a clinical pharmacologist or PharmD with specific knowledge in pharmacogenetics when in doubt before adjusting the drug dosage.For a psychotropic drug where no dose recommendation is provided consider a switch to a different medicine form from a similar drug class, when there is a genetic variant in a CYP enzyme that is involved in the metabolization.

### 3.3. Specific Recommendations per Drug Class

#### Antidepressants

Consider genotyping:In patients experiencing side effects or lack of efficacy after treatment with an adequate dose of an SSRI or SNRI. Genotyping should be considered especially in patients that experienced side effects or inefficacy with multiple psychotropic drugs with a similar CYP metabolism.In patients experiencing side effects or unexplained high or low blood drug levels in patients using TCA (tricyclic antidepressants).When next to above, there are side effects and/or inefficacy with other (somatic) pharmaca with similar CYP metabolism.

## 4. Conclusions

Pharmacogenetics is rarely used in clinical settings other than in oncology, because it is difficult to translate the results to complex real-life patient settings; people often suffer from multiple disorders and are therefore taking combinations of medications, which can influence the metabolism of the CNS active drugs. This phenomenon is called phenoconversion: conversion of e.g., extensive metabolizers into phenotypic intermediate or even poor metabolizers owing to the effects of concomitant medications on enzyme inhibition and induction, thereby modifying clinical response to drugs. This has mostly not been taken into account in clinical studies and there is a real risk that genotype-focused studies may therefore have missed clinically strong pharmacogenetic associations [6].

Recently, in the UK Biobanks PGx allele and phenotype frequencies for 487, 409 participants were analyzed, including *CYP2C19* and *CYP2D6*. For 14 CPIC pharmacogenes known to influence human drug response, it was found that 99.5% of individuals may have an atypical response to at least one drug; on average they may have an atypical response to 10.3 drugs. Nearly 24% of participants had been prescribed a drug for which they are predicted to have an atypical response [31].

A recently published article was that of Jesse Swen and colleagues, which looked at all types of medications and the impact genotyping has on side-effects [32]. The U-PGx consortium found that in n = 6944 patients, genotype guided treatment significantly reduced the incidence of clinically relevant adverse drug reactions (OR 0·70 [95% CI 0·61–0·79]; *p* < 0·0001). However, this result seems to be lower when only looking at antidepressants.

## 5. Future Perspective

Although the scientific evidence for the *CYP2C19* and *CYP2D6* genes seems strong enough to recommend clinical application, larger international and non-industry-funded implementation studies demonstrating feasibility and cost-effectiveness in real-world settings are lacking [6]. Real-world settings include people that use more than one drug, for instance, which is something that the current guidelines do not yet cover. Possible obstacles in terms of feasibility include the availability of an efficient system to generate, deliver and implement genotyping in the clinical prescription of psychiatric medication. At this time, despite gene-dosing advice for antidepressants (SSRIs, SNRIs and TCAs) as well as for many antipsychotics, these are all prescribed to patients with almost no genotyping during treatment, not even when side effects occur or a psychopharmacotherapy is inefficacious [6]. It is therefore recommended to consider genotyping for *CYP2D6* and *CYP2C19*, when there is an indication (side effects or inefficacy). Preemptive genotyping is therefore not recommended yet for psychotropic drugs [7]. Large randomized clinical trials in psychiatric patients across the world will deliver more results on how to best implement pharmacogenetics in psychiatric care.

## Data Availability

No new data were created or analyzed in this study. Data sharing is not applicable to this article.

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
