# Peer review of "Pharmacogenomics and the Management of Mood Disorders—A Review"

_jpm, 2023, doi:10.3390/jpm13071183_

Round 1

Reviewer 1 Report

The review is not well-structured and the abstract did not adequately provide guidance on the perspective or scope which the review hopes to provide.

Authors, did not provide adequate information on the following

-To introduce and provide information on current/previous treatments (drugs) used for treating depression.

- To provide an introduction on the different known metabolizing enzymes with a focus on those that are known to metabolize specific anti-depressant drug   

- No justification on the focus of CY2D6 and CYP2C19

- Case studies (If available) in countries were pharmacogenomics principles (genotype-phenotype) have been used for treatment/prescription of depression

-If possible known or well characterized genotypes of metabolizing enzymes (CY2D6 and CYP2C19) with their predicted phenotype.

-Discuss the ambiguity in genotype calling/classification of CYP2D6 variants for a predictive phenotype. Also, discuss the reclassification of some known CP2D6 genotype into a new metabolizer status which may affect implementation of pharmacogenomics in the treatment of depression

- Also, discuss the classification of metabolizer status and the predictive outcome, which is dependent on whether the drug taken is a pro-drug or already in the active form

-Also discuss limitations in implementation of pharmacogenomics such as genetic diversity in other population like those of African ancestry as well as the scarcity of pharmacogenomic/pharmacogenetics data from the African population

- For all the studies the authors have reported on, they should provide a perspective for future directives based on the reported   findings

Author Response

The review is not well-structured and the abstract did not adequately provide guidance on the perspective or scope which the review hopes to provide.

Thanks for this comment, we adjusted the manuscript and abstract accordingly and added more structure to the overall article.

Authors, did not provide adequate information on the following

-To introduce and provide information on current/previous treatments (drugs) used for treating depression.

We added a sentence in the manuscript regarding this

- To provide an introduction on the different known metabolizing enzymes with a focus on those that are known to metabolize specific anti-depressant drug   

The only well established genes that are known to influence clinical outcome of antidepressant medications are CYP2C19 and CYP2D6. We specified this in the adjusted manuscript.

- No justification on the focus of CY2D6 and CYP2C19

See above.

- Case studies (If available) in countries were pharmacogenomics principles (genotype-phenotype) have been used for treatment/prescription of depression

We have added a sentence on the countries where pharmacogenomics have been implemented in clinical care.

-If possible known or well characterized genotypes of metabolizing enzymes (CY2D6 and CYP2C19) with their predicted phenotype.

This has been described previously by groups as the Dutch Pharmacogenetic Working group and we reference the published articles on SSRIs and antipsychotics where these are described (Brouwer e.a. EJHG 2022 and Beuk e.a. EJHG 2023).We added these in the text and refs listing.

-Discuss the ambiguity in genotype calling/classification of CYP2D6 variants for a predictive phenotype. Also, discuss the reclassification of some known CP2D6 genotype into a new metabolizer status which may affect implementation of pharmacogenomics in the treatment of depression.

Although there are some different view points between different expert groups as DPWG and CPIC, the clinical implications are not known. Therefore we left this out of this manuscript. However, it would be a good idea to write a different manuscript on this. Thanks for this suggestion. We added a sentence in the revised manuscript.

- Also, discuss the classification of metabolizer status and the predictive outcome, which is dependent on whether the drug taken is a pro-drug or already in the active form.

We added a small paragraph to describe this in the revised manuscript in the introduction including some examples. Thank you for the tip

-Also discuss limitations in implementation of pharmacogenomics such as genetic diversity in other population like those of African ancestry as well as the scarcity of pharmacogenomic/pharmacogenetics data from the African population.

We added a sentence on ethnicity, which is an area that so far has been neglected in clinical studies.

- For all the studies the authors have reported on, they should provide a perspective for future directives based on the reported   findings

Thanks very much, we adjusted accordingly.

Reviewer 2 Report

After reading the manuscript with the title “Pharmacogenomics and the management of mood disorders” in depth, its structure and aim are not evident to me. It is not clear if it is a research article, a review of the literature or a clinical case.

Not knowing what the objective of the article is, it is very difficult to be able to do a critical review of the manuscript.

My suggestion is to return the manuscript as a systematic review of the literature, or as a clinical case since they report one at the end.

In any case, I will make some comments about it below.

Major comments:

1. The structure and sections of the manuscript do not comply with the regulations of the journal. The manuscript is sending like “Perspective” and in the “Article types” (https://www.mdpi.com/about/article_types) says: “Perspectives are usually an invited type of article that showcase current developments …... The structure is similar to a review, with a suggested minimum word count of 3500 words”, However, this manuscript does not have the structure of a review, according to the journal's instructions (https://www.mdpi.com/journal/jpm/instructions).

2. Point 4 of the manuscript "Dosing guidelines" could perfectly well be the introduction section or part of it since the end of the manuscript is not well understood.

3. How have the studies in Table 1 been selected and what are they discussed in the manuscript? What were the inclusion criteria, keywords used, years of studies, in which database were found, etc...? It is impossible to be able to do a critical review of the literature if that search cannot be verified.

Minor comments:

4. Lines 62-63: Where did the authors get this statement from?

5. Line 80-81: The PMs do not have activity since they have variants that give rise to the non-synthesis of the enzyme or to a non-functional enzyme, so the term "little" is inaccurate

6. All gene names should always be in italics.

Author Response

After reading the manuscript with the title “Pharmacogenomics and the management of mood disorders” in depth, its structure and aim are not evident to me. It is not clear if it is a research article, a review of the literature or a clinical case.

Not knowing what the objective of the article is, it is very difficult to be able to do a critical review of the manuscript.

My suggestion is to return the manuscript as a systematic review of the literature, or as a clinical case since they report one at the end.

In any case, I will make some comments about it below.

Thanks for your helpful suggestion, we now focused the article on the review and took out the case history.

Major comments:

  1. The structure and sections of the manuscript do not comply with the regulations of the journal. The manuscript is sending like “Perspective” and in the “Article types” (https://www.mdpi.com/about/article_types) says: “Perspectives are usually an invited type of article that showcase current developments …... The structure is similar to a review,with a suggested minimum word count of 3500 words”, However, this manuscript does not have the structure of a review, according to the journal's instructions (https://www.mdpi.com/journal/jpm/instructions).

We adjusted the article accordingly

  1. Point 4 of the manuscript "Dosing guidelines" could perfectly well be the introduction section or part of it since the end of the manuscript is not well understood.

Thanks for this wonderful suggestion, we adjusted accordingly.

  1. How have the studies in Table 1 been selected and what are they discussed in the manuscript? What were the inclusion criteria, keywords used, years of studies, in which database were found, etc...? It is impossible to be able to do a critical review of the literature if that search cannot be verified.

See the already published search strategy at van Westrhenen e.a. Frontiers 2021. We have referenced this now clearly in the revised manuscript and added a small method section.

Minor comments:

  1. Lines 62-63: Where did the authors get this statement from?

Earlier published, added reference

  1. Line 80-81: The PMs do not have activity since they have variants that give rise to the non-synthesis of the enzyme or to a non-functional enzyme, so the term "little" is inaccurate.

Thanks you for your comment. Poor metabolizer refers to people that have inactive but also they can have low active alleles, therefore we always use the term little. We adjusted to clarify.

  1. All gene names should always be in italics

We adjusted this in the revised article

Round 2

Reviewer 1 Report

None

Reviewer 2 Report

The authors adequately respond to previous comments and have also modified the manuscript accordingly.
However, you must review and correct the entire list of references since not all of them have the same format. You must check the format of the journal and change all references according to it.